# Actinide Ion (Americium-241 and Uranium-232) Interaction with Hybrid Silica–Hyperbranched Poly(ethylene imine) Nanoparticles and Xerogels

**DOI:** 10.3390/gels9090690

**Published:** 2023-08-27

**Authors:** Ioannis Ioannidis, Ioannis Pashalidis, Michael Arkas

**Affiliations:** 1Laboratory of Radioanalytical and Environmental Chemistry, Department of Chemistry, University of Cyprus, P.O. Box 20537, Cy-1678 Nicosia, Cyprus; ioannides.c.ioannis@ucy.ac.cy; 2National Centre for Scientific Research “Demokritos”, Institute of Nanoscience and Nanotechnology, 15310 Athens, Greece

**Keywords:** Am-241 and U-232, dendritic polymers, silica xerogels, composites, dendrimers, water purification, radioactive wastewater, radionuclide removal, thermodynamic, environmental remediation, water decontamination

## Abstract

The binding of actinide ions (Am(III) and U(VI)) in aqueous solutions by hybrid silica–hyperbranched poly(ethylene imine) nanoparticles (NPs) and xerogels (XGs) has been studied by means of batch experiments at different pH values (4, 7, and 9) under ambient atmospheric conditions. Both materials present relatively high removal efficiency at pH 4 and pH 7 (>70%) for Am(III) and U(VI). The lower removal efficiency for the nanoparticles is basically associated with the compact structure of the nanoparticles and the lower permeability and access to active amine groups compared to xerogels, and the negative charge of the radionuclide species is formed under alkaline conditions (e.g., UO_2_(CO_3_)_3_^4−^ and Am(CO_3_)_2_^−^). Generally, the adsorption process is relatively slow due to the very low radionuclide concentrations used in the study and is basically governed by the actinide diffusion from the aqueous phase to the solid surface. On the other hand, adsorption is favored with increasing temperature, assuming that the reaction is endothermic and entropy-driven, which is associated with increasing randomness at the solid–liquid interphase upon actinide adsorption. To the best of our knowledge, this is the first study on hybrid silica–hyperbranched poly(ethylene imine) nanoparticle and xerogel materials used as adsorbents for americium and uranium at ultra-trace levels. Compared to other adsorbent materials used for binding americium and uranium ions, both materials show far higher binding efficiency. Xerogels could remove both actinides even from seawater by almost 90%, whereas nanoparticles could remove uranium by 80% and americium by 70%. The above, along with their simple derivatization to increase the selectivity towards a specific radionuclide and their easy processing to be included in separation technologies, could make these materials attractive candidates for the treatment of radionuclide/actinide-contaminated water.

## 1. Introduction

Americium (Am) and uranium (U) are two elements belonging to the actinide series [1]. These elements exhibit different species when subjected to different pH environments and oxidation states. In aqueous solutions, uranium forms various hydrolysis and carbonate complexes, such as UO_2_^2+^ in acidic conditions, UO_2_(OH)_2_ and UO_2_OH^+^ near neutral pH, and UO_2_(CO_3_)_2_^2−^, UO_2_(CO_3_)_3_^4−^ in alkaline environments [2,3]. On the other hand, for americium, Am^3+^ dominates in acidic aqueous solutions and under normal conditions, Am(III)-hydroxycarbonate (Am(CO_3_)(OH)) shows limited solubility, and Am(CO_3_)_3_^3−^ is the dominant species in alkaline aqueous environments [4,5]. In the literature, there are only a few studies that report on the adsorption of these two radionuclides by various adsorbent materials such as microplastics [6] and aerogels [7]. According to those studies, the adsorption mechanism seems to depend on both the radionuclide speciation and the surface-active sites, including charge/polarization and complex formation properties.

Due to the severity of the continuously increasing accumulation of these radioactive ions in aquatic media, there has been extensive research on appropriate methods such as cation exchange, precipitation, and extraction to address the problem. Adsorption, though, is established as the most effective. The incorporation of radionuclides into the substrates proceeds mainly via two mechanisms: (1) electrostatic attraction between the solubilized ion forms and oppositely charged adsorbents producing outer sphere complexes or (2) formation of coordination compounds with the terminal functional groups of the adsorbents (e.g., -COOH, -NH_2_, -OH) yielding inner-sphere complexes [8,9]. Effective and specific pollutant retention materials were discovered to demonstrate extraordinary properties. More specifically, from a purely organic point of view, microplastics [10], biopolymers [11], and conventional polymers [12] are the more promising representatives while distinguished classes of inorganic compounds include metal oxides [13], minerals [14], and carbon allotropes [15]. The combination of organic and inorganic moieties expectedly yields a wider variety of prominent adsorbents such as, for instance, metal–organic frameworks (MOFs) [16], aerogels [17,18,19,20], organic–inorganic composites [21,22], and hybrids [23,24]. 

The applicability of dendritic polymers [25,26,27] in water purification is nowadays well established [28]. This fourth class of macromolecular architecture is generated by the replacement of the typical linear polymerization, branching, or crosslinking with radial propagation of the monomeric units [29]. Because of their noteworthy, branched architecture, reminiscent of the tree-like structures commonly encountered in nature (roots and branches of trees, arteries, and river deltas [30]), they may adapt to almost all conventional pollutant removal methods. For this reason, there is a wealth of literature on adsorption [31], catalytic degradation [32], and ultrafiltration [33] procedures. The efficacity of all main dendritic classes relies on the three main structural components that define their architecture: (a) The core or focal point in the case of dendrons; (b) the internal containing cavities formed by the polymerized units: the “branches” of the “tree”. They define the adsorption properties of the whole molecule and may be modified accordingly to incorporate the majority of pollutants; (c) the peripheral groups that may undergo functionalization to fine-tune solubility, electrostatic interactions, and any other desired property.

Since adsorption proved the more prosperous method for aquatic systems remediation from industrial radioactivity pollution, research on the performance of dendritic polymers in this particular domain attracted analogous interest that led to the development of specialized solutions [34,35,36,37]. To extend the proof of concept to an economically feasible implementation, the first requirement is diminishing the cost of the dendritic polymer. An obvious approach is to replace the expensive dendrimers [38,39,40,41,42] with their non-symmetric counterparts, hyperbranched polymers [43,44,45,46,47,48,49]. Their properties and overall behavior are similar, and they have already taken the place of their monodisperse counterparts in many implementations in diagnostics [50,51,52], biosensors [53,54], antimicrobial protection [55], liquid crystals [56,57], and drug delivery [58].

A second necessity is a supporting substrate that protects the organic moieties from chemical degradation and preferably also possesses pollutant retention capacity or other separation capabilities such as magnetic susceptibility. The combination with porous inorganic materials [59] is the most obvious formulation. There are three main approaches in this direction: (a) simple impregnation of the carrier with a solution of the hyperbranched polymer [60]; (b) chemical grafting of the dendritic polymer to the substrate [61,62]; (c) immobilization of a functionalized derivative by polymerization into the pores of the supporting scaffold [63]. 

A fourth perspective that addresses both ecological and economic issues is based on the ability of hyperbranched poly(ethylene imine) (PEI) to mimic the action of silaffins and produce silica nanospheres [64]. Alternatively, the same molecules may intervene in the hydrogen bond network that forms during the gelation of orthosilicic acid and thus produce xerogels after drying. Both silica–PEI composites exhibited potent adsorbing capability for many aqueous pollutants [65,66]. The two synthesis procedures are very simple and take place at room temperature with no involvement of toxic organic solvents or byproducts. The present work aims to test these two species’ (nanospheres and xerogels) radionuclide adsorption kinetics and thermodynamics. The binding efficiency has been evaluated in terms of the linear adsorption constant K_d_ and the relative removal efficiency. The study will be performed for factors such as pH and temperature for ultra-trace level as well as in seawater to define the optimal uranium and americium removal conditions. Although other studies on the removal of uranium, thorium [67], and other radionuclides have been reported [68], to the best of our knowledge, this is the first study on hybrid silica–hyperbranched poly(ethylene imine) nanoparticle and xerogel materials used as adsorbents for americium and uranium present in the studied solutions at ultra-trace levels (about nine orders of magnitude lower actinide concentration). We have also included americium in the present study, because americium is present, like other trans-uranium elements at very low levels, in marine environments because of global fallout from nuclear bomb tests, accidental releases (e.g., the Fukushima accident), and effluent discharges from irradiated fuel reprocessing [69,70,71]. However, studies dealing with americium removal from radioactively contaminated waters are limited and mainly associated with the use of inorganic adsorbents/exchangers from intermediate (waste)water solutions [68,72].

## 2. Results and Discussion

### 2.1. Xerogel and Nanoparticle Characteristics

As established in the experimental section, the two different forms of hybrid silica–PEI composites arise by modification of the orthosilicic acid/hyperbranched PEI ratio. The most profound difference thus is the dendritic polymer content of the nanoparticles, which is more than double that of the xerogel [66]. The higher number of positively charged amino groups has a secondary consequence. The external charge of both adsorbents at pH 3 is similar. Their behavior though by increasing pH is completely disparate. The nanoparticles remain positive even at pH 9, albeit of a much lesser charge. Their behavior is very similar to that previously observed in a detailed investigation performed for analogous nanoparticles formed by the mediation of hyperbranched PEI with Mw 5000 [65]. In contrast, an abrupt reverse is observed for the xerogels that became negative even at pH 4 (Figure 1). It is important to note that the ζ potential values of xerogels are in between those of the hyperbranched-PEI silica nanoparticles and those of pure silica nanoparticles that are slightly positive at pH 2 and became negative at pH 3. 

Besides the composition, the role of hyperbranched PEI is different in the creation of the final hybrid material. In the case of nanoparticles, the dendritic polymer functions as a catalyst and matrix and is positioned in the core of the composite enveloped by a shell of the produced silica. In the case of gels, there are no indications that PEI promotes gelation. Xerogel conformation has much lower anisotropy and consists both in the interior and the exterior of “islands” of dendritic polymers surrounded by a “sea” of siloxane and silanol groups. In this way, the cavities of the adsorbing polymer are more exposed to the radionuclides. Another significant variance is the pore structure. The xerogels have larger BET surfaces but much smaller pores (BET surface for PEI 750,000 nanoparticles is 271.4 m^2^/g; BET surface for PEI 750,000 xerogel is 139.0 m^2^/g; mean pore size d_mean_ as 4000·TPV/S_BET_ is 6.8 nm for xerogel and 18.4 nm for nanoparticles) [66]. The most important advantage of xerogels, though, is their capability to be readily formed in restricted media such as, for instance, the pores of a ceramic support by simple immersion of the latter into a precursor gel solution. Thus, in addition to the classical powder formulation that may be dispersed into the contaminated water, there is also the option of coatings applied to continuous filtration filters. Their stability will be secured due to the in situ polymerization. 

Based on the above, the binding of the actinides by the nanoparticles and xerogels is expected to occur via cation exchange between protons and the U(VI) or Am(III) cations and inner-sphere complex formation between the actinide cations and the polyimine moieties. Ion-exchange and inner-sphere complex formation occur successively, particularly in the acidic pH region, where both studied actinides (Am(III) and U(VI)) are expected to be positively charged. Specifically, when the actinide cations approach the protonated imine groups, the proton is exchanged by the actinide ion (ion exchange), which is then complexed by the imine groups through the interaction between the amine lone pair and the empty actinide orbitals. In the neutral and alkaline pH region (due to the extremely low actinide concentration), the actinide complexation may occur directly with the partially de-protonated amino groups [67]. In the case of uranium (e.g., uranyl moiety), interaction of the uranyl oxygens and ammonium protons is also possible, favoring surface binding of U(VI) (Figure 2) [6,7].

### 2.2. Adsorption Kinetics

The time evolution of the relative actinide ion binding at pH 4, pH 7, and pH 9 is graphically presented in Figure 3a–c. It is clear that the adsorption kinetics differ significantly between the different adsorbents (e.g., nanoparticles and xerogels) and actinide ions (e.g., U(VI) and Am(III)) and that pH plays a main role since it determines both the charge of the surface and the actinide speciation in solution. According to the kinetic data, which have been obtained for the adsorption of uranium and americium by the nanoparticle and xerogel adsorbents at three different pH regions, generally, uranium presents higher adsorption rates compared to americium. Regarding the adsorbents, under acidic conditions (pH 4) the nanoparticles present higher adsorption kinetics for both radionuclides, whereas at pH 9 the xerogels show higher adsorption kinetics than the nanoparticles. Under neutral conditions, the adsorption kinetics are almost similar, except for the adsorption of americium by the nanoparticles, which present lower adsorption kinetics. Moreover, the experimental kinetic data are well described by the Lagergren equation (e.g., the pseudo-first-order kinetic model), indicating that the binding rate is dependent only on the actinide concentration, which is expected for systems performed at ultra-trace levels [73]. Following, the thermodynamic experiments were evaluated after 10 days of contact time in order to assure equilibrium conditions.

### 2.3. pH Effect on the Actinide Binding Efficiency

The % relative actinide ion removal by the nanoparticle and xerogel adsorbents is graphically presented in Figure 4 and indicates that in the case of xerogels, the pH does not significantly affect the removal efficiency. On the other hand, the nanoparticle affinity towards both radionuclides is considerably affected by pH. Specifically, the % relative removal decreases with increasing pH, and at pH 9 the values of the relative removal are below 50%. To understand this behavior, we must take into account the predominant radionuclide species at each pH and their charges in conjunction with those of the adsorbents [7]. At low pH levels, the predominant ion species (Am^3+^, UO_2_^2+^) are positively charged and electrostatically rappelled by the homonymous charges of both adsorbents. Yet, their removal from the aqueous solution proceeds smoothly and is almost complete in the case of nanoparticles that exhibit a higher positive charge. This is a strong indication that adsorption proceeds mainly via the chelation mechanism affording inner-sphere complexes. At neutral pH, the americium ions are positive (Am^3+^, AmCO_3_^+^), justifying a small improvement in their incorporation into the negatively charged hydrogels. Uranium counterparts are neutral (UO_2_CO_3_) or positive (UO_2_OH^+^), and a possible explanation for the slight drop in the adsorption must be sought to the larger ion sizes and the core–shell structure of the nanoparticles. It seems that the bigger species are struggling to pass through the porous silica entourage to form coordination compounds with the hyperbranched PEI core. This effect becomes more intense in basic solutions where the negatively charged and much bulkier actinide species (UO_2_(CO_3_)_3_^4−^, Am(CO_3_)_2_^−^ present a slightly lower affinity to the homonymous xerogels by repelling one another and experience a far greater steric hindrance when crossing the silica periphery of the nanoparticles, resulting in considerably lower removal efficiency. Moreover, the data indicate that both materials present a higher affinity for U(VI) compared to Am(III). Compared to alginate aerogels, the xerogels present significantly higher removal efficiency for both uranium and americium [7]. Similarly, the nanoparticles show higher relative removal efficiencies, except for pH 9, where the relative removal declines to values below 50%.

Regarding the K_d_ values (Figure 5), the highest values have been determined for the U(VI) adsorption by XG, which reaches a maximum value in neutral solutions. This behavior is attributed to the competitive reaction of protons in the acidic pH region, which occupy binding sites on the composite surface and the UO_2_(CO_3_)_3_^4−^ species, the formation of which is favored in the alkaline pH region (pH 9) stabilizing U(VI) in solution [74]. On the other hand, for the nanoparticles that present generally lower K_d_ values, the maximum K_d_ value is observed in the weak acidic pH region (pH 4) and drops dramatically with increasing pH. Regarding Am(III), the associated K_d_ values present similar behavior with the corresponding K_d_ values of U(VI). However, the associated K_d_ values of Am(III) are significantly lower than the corresponding K_d_ values of U(VI). The higher affinity of the xerogels and nanoparticles for U(VI) is of particular interest regarding uranium separation from trivalent actinide- and lanthanide-containing solutions. However, the associated K_d_ values are orders of magnitude lower, indicating lower chemical affinity of the xerogels and nanoparticles for the studied actinides compared to the alginate aerogels. This could be associated with the lower density of the alginate aerogels but also with the fact that the PEI xerogels and nanoparticles have a larger number of binding sites available for actinide ion complexation compared to the alginate aerogels [5,22].

The removal of actinides (including americium and uranium) from low and intermediate active solutions has been studied using inorganic adsorbents/exchangers (e.g., titanosilicates, layered manganese oxides, iron and titanium oxides, and nano-cerium vanadate [71,75,76,77]. The associated adsorption kinetics are significantly faster and the adsorption efficiencies (% removal and K_d_ values) remarkably higher. However, both the faster kinetics and the higher adsorption efficiencies could be ascribed to the significantly higher actinide concentrations used in those studies, which are more than several orders of magnitude higher than the uranium and americium levels used in the present study.

### 2.4. Adsorption Thermodynamics

The temperature effect on the relative actinide ion removal is shown in Figure 6 and Figure 7 for uranium and americium, respectively. The temperature increase from 25 °C to 35 °C generally results in an increase in the removal efficiency, indicating that the binding is favored with increasing temperature and that the entropy governs the adsorption process. However, at 45 °C the removal efficiency declines significantly due to changes in the structure of the adsorbents, which is associated with solvent molecule release at the given temperature. A similar effect was observed also in a previous study investigating the U(VI) adsorption by the same type of adsorbents at increased uranium concentrations [66]. Hence, the thermodynamic parameters ΔH° and ΔS° have been evaluated for the three different pH areas using the experimental data obtained at 25, 30, 35, and 40 °C, and the associated lnK_d_ vs. 1/T are presented in the Appendix A.

The thermodynamic parameters ΔH° and ΔS° for the actinide ion adsorption by the xerogels and nanoparticles at pH 4, pH 7, and pH 9 have been evaluated using the linear form of the Van’t Hoff formula, and the associated data are graphically shown in Figure 8 and Figure 9, respectively. According to the data in Figure 8, the ΔH^o^ values are positive, suggesting an endothermic process. The lowest ΔH^o^ values for uranium are observed in the acidic pH area (pH 4) and the highest in the neutral pH area (pH 7) for the xerogels and in the alkaline area (pH 9) for the nanoparticles. For americium, the lowest ΔH° values have been determined at pH 7 and pH 4, for the xerogels and the nanoparticles, respectively. On the other hand, the highest ΔH° values for americium have been determined at pH 4 and pH 9 for the xerogels and the nanoparticles, respectively. Moreover, the nanoparticles present significantly higher ΔH° values for both radionuclides at pH 9. The latter is most probably associated with the formation of negatively charged species (e.g., UO_2_(CO_3_)_3_^4−^ and Am(CO_3_)_2_^−^) in the alkaline pH range, which are extensively stabilized in solution. The endothermic character of the U(VI) surface binding at ultra-trace radionuclide levels has also been observed for the U(VI) binding by oxidized biochar fibers at pH 4 [73]. However, in the case of Am(III) binding by oxidized biochar fibers at pH 4, the adsorption process was exothermic and close to the values observed for the Am(III) adsorption by nanoparticles at pH 4.

### 2.5. Radionuclide Interaction in Seawater

The removal efficiency of the americium and uranium radionuclides from seawater has been investigated after tracing seawater samples with americium-241 and uranium-232 and contacting them with the NP and XG adsorbents. The associated data are shown in Figure 10 and point out that, in the case of XG, the adsorption efficiency for the two actinides is similar (~90%). On the other hand, the NP adsorbents present slightly higher adsorption efficiency for uranium (~80%) compared to americium (70%).

The higher affinity of the adsorbents is depicted in Figure 11, which summarizes the K_d_ values of NPs and XGs for uranium and americium. According to the data in Figure 10, the XGs possess higher chemical affinity for both radionuclides compared to the NP adsorbents, and the latter present significantly higher affinity for uranium compared to americium. This effect is similar to and has been observed also in laboratory/de-ionized water solutions. Nevertheless, despite the relatively low K_d_ values, both materials present far higher removal efficiency of americium and uranium from seawater than oxidized biochar [72] or even X-alginate aerogels [7]. Xerogels could remove both actinide isotopes from seawater by almost 90%, whereas nanoparticles could remove uranium by 80% and americium by 70%. The above, along with their simple derivatization to increase the selectivity towards a specific radionuclide and their easy processing to be included in separation technologies, could make these materials very attractive for the treatment of radionuclide/actinide-contaminated waters.

## 3. Conclusions

The conclusions that can be drawn from this study are the following:Hybrid silica–hyperbranched poly(ethylene imine) nanoparticles and xerogels present relatively high removal efficiency at pH 4 and pH 7 (>70%) for Am(III) and U(VI).Generally, the adsorption process is relatively slow due to very low radionuclide concentrations and is governed by the actinide diffusion from the bulk solution to the composite surface.The actinide binding by the NP and XG composites is favored by increasing temperature indicating an endothermic and entropy-driven binding reaction.Compared to other adsorbents, which have been investigated regarding the removal of the studied actinide ions, both composites show far higher removal efficiency from laboratory and seawater samples, which is for xerogels almost 90% and for nanoparticles about 80% for uranium and 70% for americium.The simple derivatization of NPs and XGs to increase the selectivity towards specific actinides and other metal ions, along with their easy implementation in water treatment technologies, could make these materials attractive candidates for the decontamination of actinide-contaminated waters, including seawaters.

## 4. Experimental Section

### 4.1. Synthesis of the Composite Silica–PEI 750,000 Nanoparticles

A variation of the method proposed by Knecht et al. [78,79] for poly(amidoamine) PAMAM and poly(propylene imine) PPI dendrimers was employed for the preparation of the silica–PEI nanocomposites, as reported in our previous work [65]. In brief, a solution of hyperbranched PEI 750,000 (Mn = 750,000 BASF, Ludwigshafen, Germany) solution 20 mM in primary and secondary amines was prepared by dissolving 0.29 g PEI 750,000 in 250 mL 20 mM phosphate buffer pH 7.5 (K_2_HPO_4_ Carlo Erba Reagenti, Rodano, Milano, Italy, KH_2_PO_4_ Merck Darmstadt Germany). Then, 1 mL of this solution was added to 10 mL orthosilicic acid (1 M) prepared from the hydrolysis tetra ethoxy silane (Sigma-Aldrich, Steinheim, Germany) in 5 mM HNO_3_ under vigorous stirring for 15 min. Silica precipitate was collected after centrifugation (10 min 12,000× *g*), washing with water, and drying under vacuum over P_2_O_5_ (Sigma-Aldrich, Steinheim, Germany); yield 77%. 

### 4.2. Synthesis of the Silica–PEI 750,000 Xerogels

This procedure is also described in detail in previous work [80]. Similar acid hydrolysis of a 1 M tetraethoxysilane solution (Sigma-Aldrich, Steinheim, Germany) with 25 µL HNO_3_ under stirring for 15 min produced 1 M orthosilicic acid. To 5 mL of this solution, 5 mL of PEI 750,000 (BASF, Ludwigshafen, Germany) aqueous solution (40 mM in primary and secondary amine groups) was added. The pH of the gel precursor solution was adjusted to 7.5 by adding K_2_HPO_4_ powder (Carlo Erba Reagenti, Rodano, Milano, Italy). Hydrogel formation was observed after about 1 to 2 h, and the reaction product was dried overnight over phosphorus pentoxide (Sigma-Aldrich, Steinheim, Germany) under vacuum to yield the silica–PEI 750,000 xerogel. 

### 4.3. Adsorption Experiments

All adsorption experiments were conducted in 20 mL polyethylene vials under an ambient atmosphere and at various temperatures (25, 30, 35, 40, 45 °C). The test and reference solutions were prepared from the actinide standard solutions of uranium-232 (National Physical Laboratory, Teddington, UK) and americium-241 (North America Scientific Inc., Los Angeles, CA, USA) with radioactivity levels of 4.923 and 12.05 kBq/g, respectively. The composite adsorbents employed were based on dendritic poly(ethylene imine) 750,000: silica–PEI 750,000 nanoparticles (NPs) and silica–PEI 750,000 xerogels (XGs). The experiments were performed in de-ionized water solutions, and the pH was adjusted to different values (pH 4, 7, and 9) using dilute aqueous solutions of NaOH and HCl and seawater solution (SW). The seawater sample used in the present study was obtained from a local beach in Cyprus, and the radioactive solutions were prepared after the SW was filtered to remove any impurities. Radionuclides analysis was carried out using an alpha spectrometer (Canberra) by using stainless steel discs previously deposited with the two isotopes of U-232 and Am-241, as described elsewhere [81]. 

The adsorption experiments were performed in 10 mL of a mixture of americium-241 and uranium-232 with an activity concentration of 0.5 mBq/mL for each radioisotope, which corresponds to molar concentration, [U-232] = 8.6 × 10^−14^ mol/L and [Am-241] = 8.17 × 10^−13^ mol/L. A small amount of 0.005 g for each material was added to the solution, and the adsorption kinetics was studied for 10 days. The mixture in the flasks was stirred (45 min^−1^) on a shaker (SK-R1807, DLAB). At specific time intervals, a 50 μL sample was taken from the test solutions, and the actinide concentration was determined by means of alpha-spectroscopy. The alpha-spectrometer was previously calibrated by sample analysis using a standard reference source.

As the concentration of radionuclides is in the picomole range and the binding sites (B) of the composite surface are in large excess compared to the initial concentration of radionuclides, the partition coefficient, K_d_, adequately describes the equilibrium associated with the sorption of U-232 and Am-241:K_d_ = C_ads_/C_aq_ (L/Kg)(1)
where C_ads_ (Bq/g) is the activity concentration of the actinide absorbed, and C_aq_ (Bq/L) is the concentration of the actinide in solution. The amount of U-232 and Am-241 adsorbed is calculated by subtracting the number of actinides adsorbed on the walls of the polyethylene vial from the total amount of actinide adsorbed.

To calculate the standard enthalpy (ΔH°) and entropy (ΔS°) of the system, the following formula was used:(2)$$\ln K_{d}=\frac{-{\Delta H}^{o}}{RT}+\frac{{\Delta S}^{o}}{R}$$

The value of K_d_ is directly affected by both ΔH° and ΔS° and is inversely influenced by temperature. The slope of the relationship is determined by dividing the ΔH by the temperature (in Kelvin), multiplied by the gas constants (R). On the other hand, the intercept is determined by dividing the ΔS° by the R.

The experiments were carried out in duplicate, and the data evaluation was based on the mean values calculated from the results from the two different experiments. The relative uncertainty, which is basically determined by the measurement uncertainty, was estimated to be below 10%.

## Figures and Tables

**Figure 1 gels-09-00690-f001:**
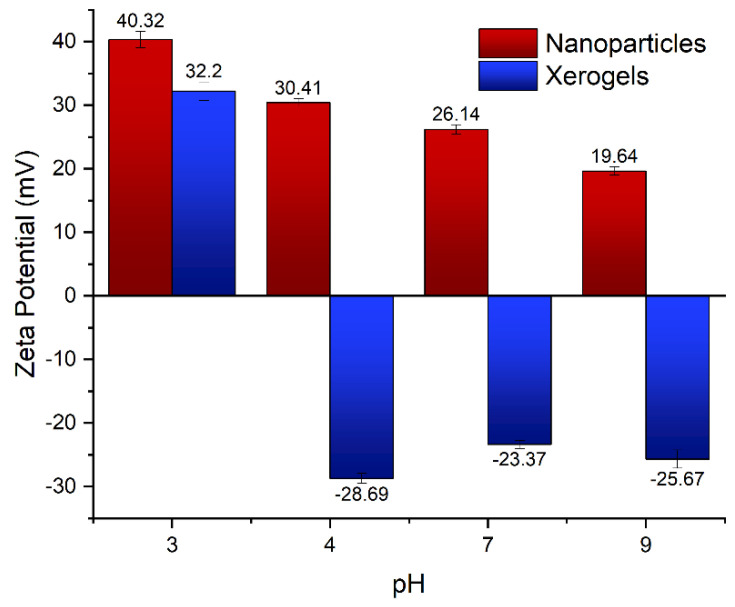
Surface charge of nanoparticles and xerogels at different pH.

**Figure 2 gels-09-00690-f002:**
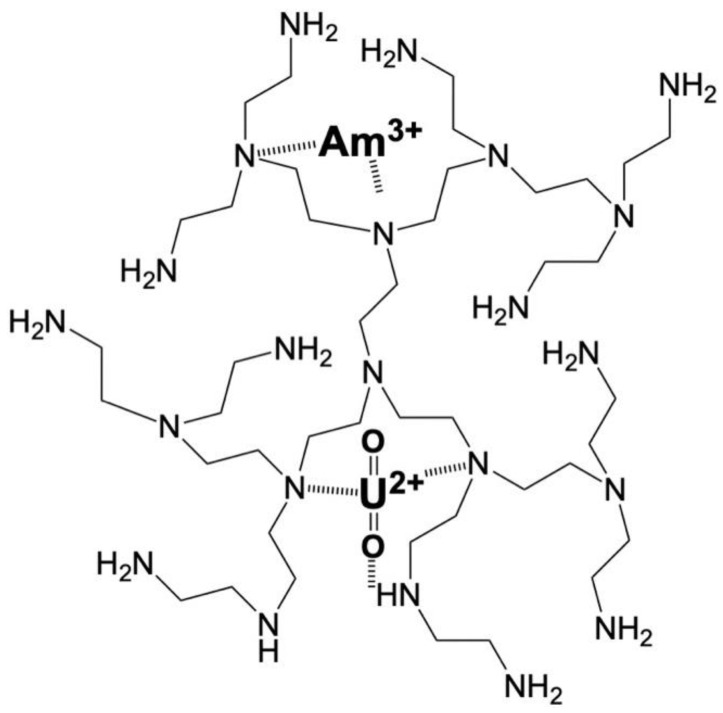
Schematic illustration of possible interactions of the actinide ions (Am^3+^ and UO_2_^2+^) with the PEI-based adsorbents.

**Figure 3 gels-09-00690-f003:**
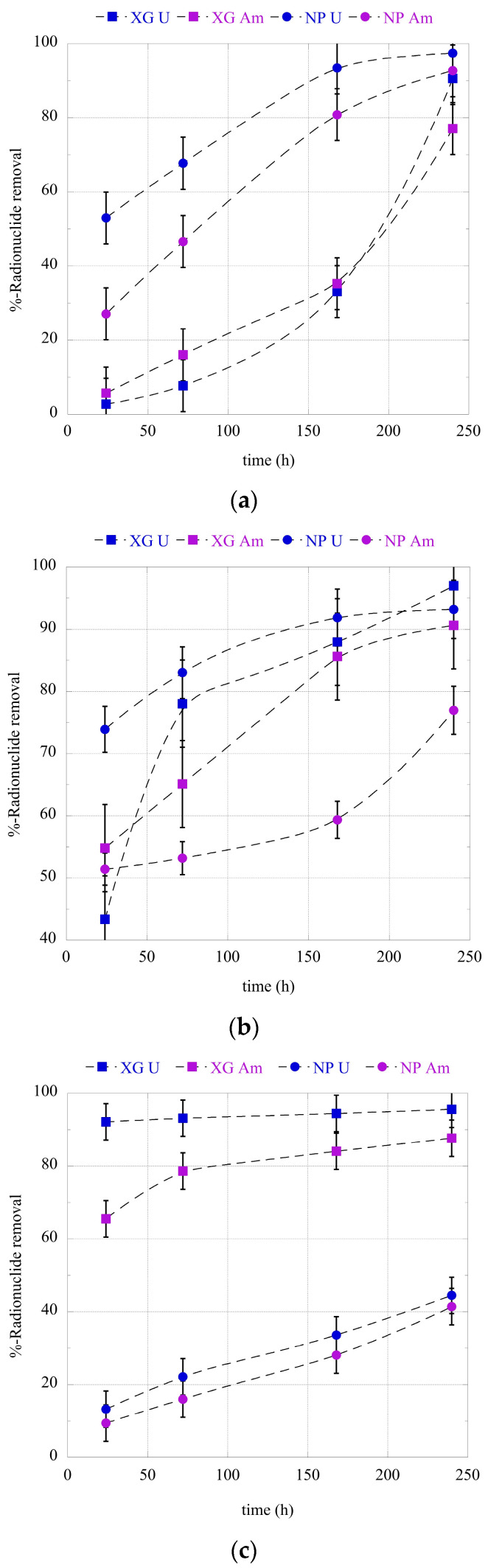
U(VI) and Am(III) adsorption kinetics by the PEI nanoparticles (NPs) and xerogels (XGs) at ultra-trace levels at (**a**) pH 4, (**b**) pH 7, and (**c**) pH 9. Experimental conditions: 0.5 Bq/mL U-232 and Am-241 in 10 mL total volume and ambient conditions.

**Figure 4 gels-09-00690-f004:**
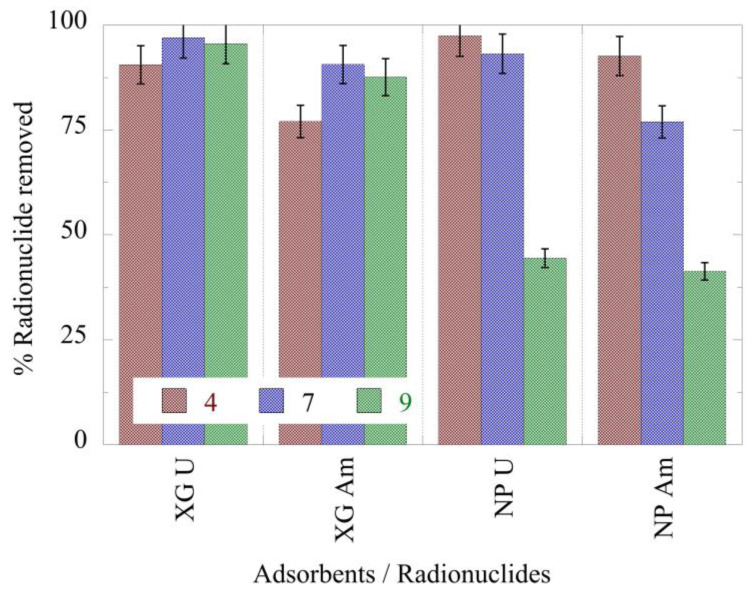
% relative actinide ion removal by nanoparticles (NPs) and xerogels (XGs) at ultra-trace levels as a function of pH from aqueous solutions. Experimental conditions: 10 mL of the solution, with 0.5 Bq/mL for both U-232 and Am-241 tracers, in different pH (pH = 4, 7, 9) and at room temperature.

**Figure 5 gels-09-00690-f005:**
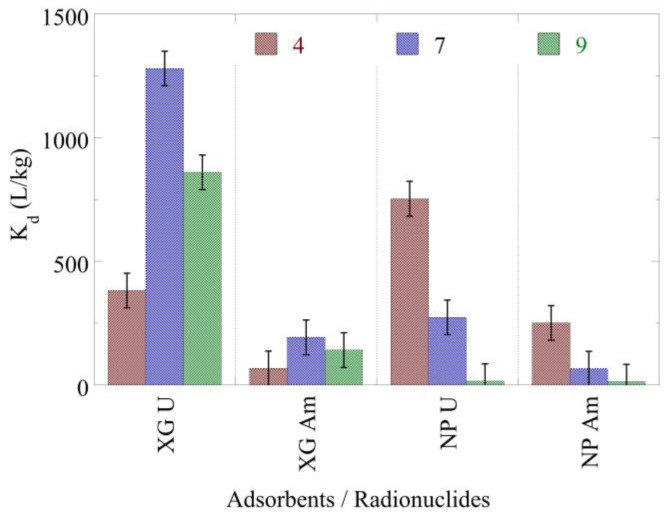
Adsorption efficiency (log_10_K_d_) of U(VI) and Am(III) by nanoparticles (NPs) and xerogels (XGs) at ultra-trace levels as a function of pH from aqueous solutions. Experimental conditions: 10 mL of the solution, with 0.5 Bq/mL for both U-232 and Am-241 tracers, in different pH (pH = 4, 7, 9) and at room temperature.

**Figure 6 gels-09-00690-f006:**
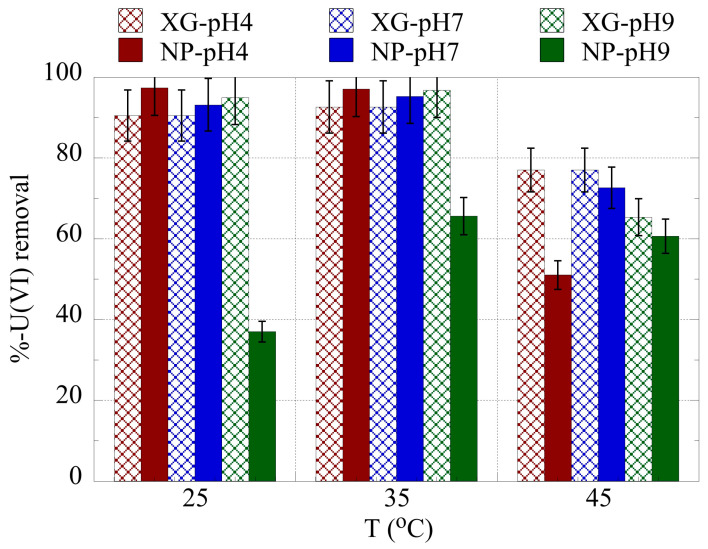
% relative removal of U(VI) by nanoparticles (NPs) and xerogels (XGs) at ultra-trace levels as a function of pH and temperature. Experimental conditions: 10 mL of the solution, with 0.5 Bq/mL for both U-232 and Am-241 tracers, in different pH (pH = 4, 7, 9) and temperatures (25, 35, 45 °C).

**Figure 7 gels-09-00690-f007:**
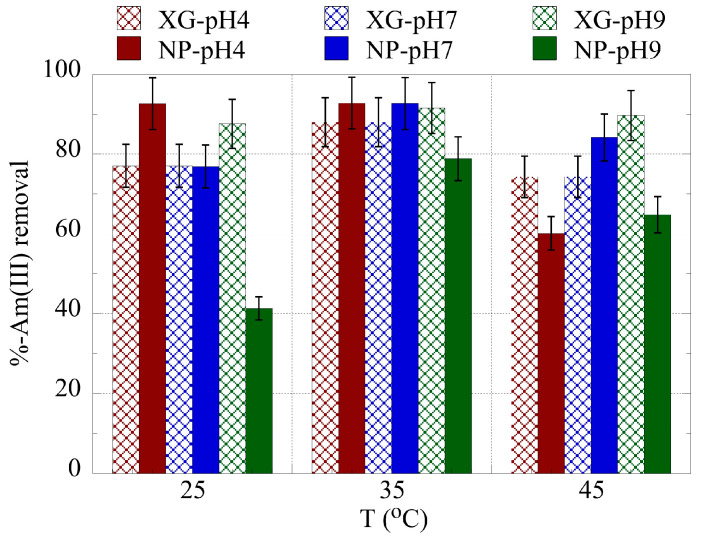
% relative removal of Am(III) by nanoparticles (NPs) and xerogels (XGs) at ultra-trace levels at pH 4, 7, and 9, as a function of temperature. Experimental conditions: 10 mL of the solution, with 0.5 Bq/mL for both U-232 and Am-241 tracers, in different pH (pH = 4, 7, 9) and temperatures (25, 35, 45 °C).

**Figure 8 gels-09-00690-f008:**
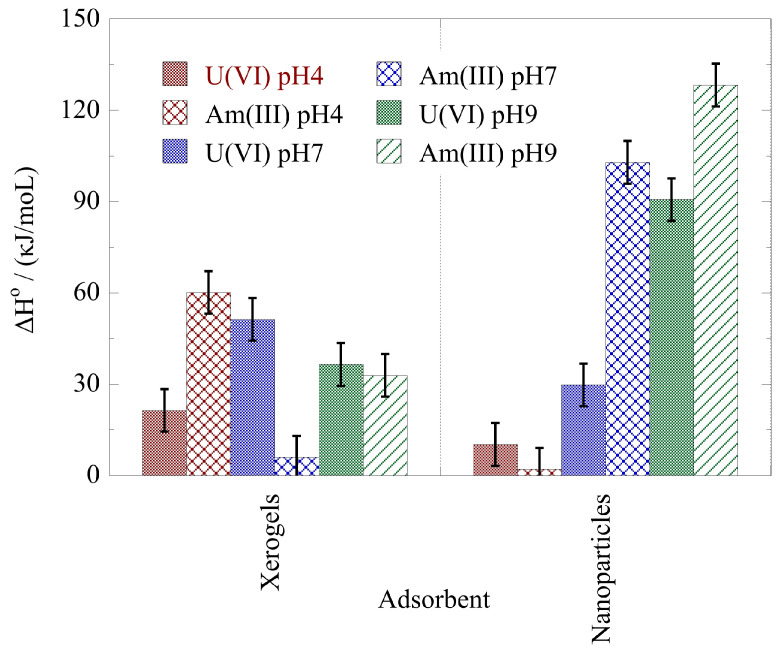
ΔH° of the actinide ion adsorption by nanoparticles (NPs) and xerogels (XGs) at ultra-trace levels at pH 4, 7, and 9. Experimental conditions: 10 mL of the solution, with 0.5 Bq/mL for both U-232 and Am-241 tracers, at different temperatures (25, 35, 45 °C).

**Figure 9 gels-09-00690-f009:**
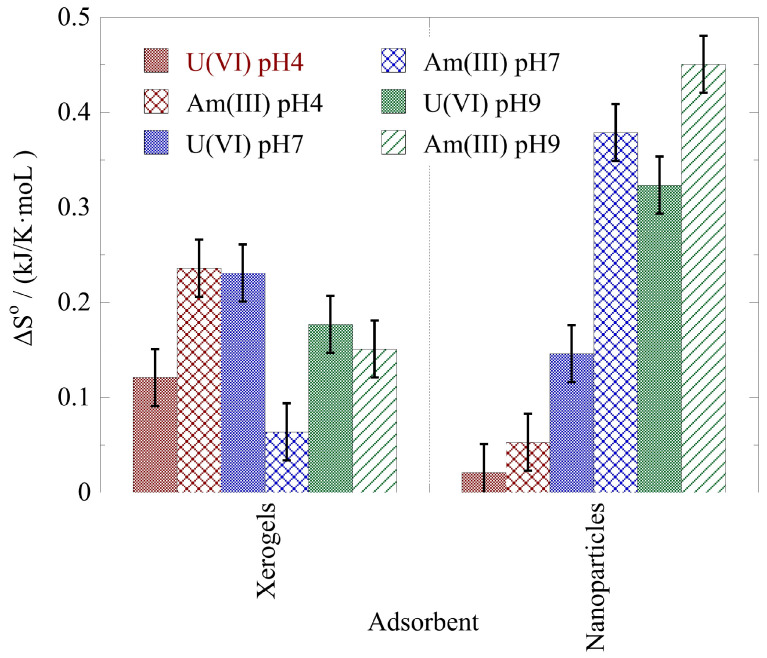
ΔS° of the actinide ion adsorption by nanoparticles (NPs) and xerogels (XGs) at ultra-trace levels at pH 4, 7, and 9. Experimental conditions: 10 mL of the solution, with 0.5 Bq/mL for both U-232 and Am-241 tracers, at different temperatures (25, 35, 45 °C).

**Figure 10 gels-09-00690-f010:**
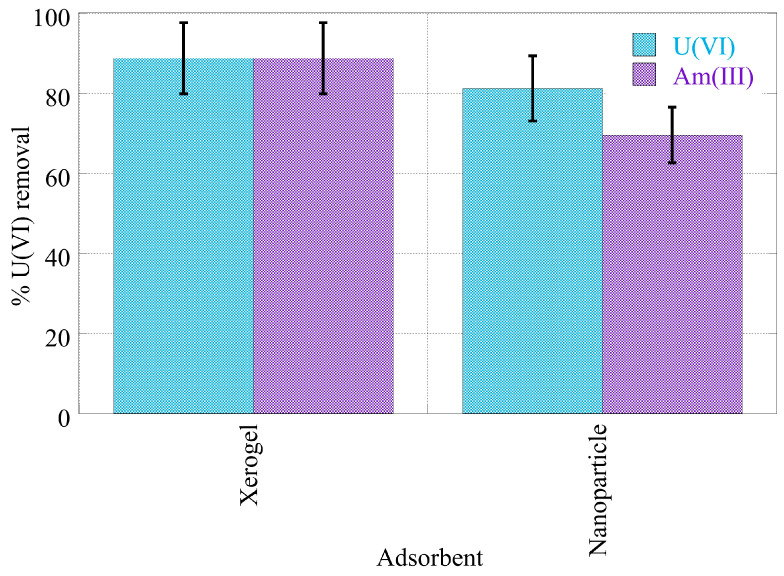
% relative U(VI) and Am(III) removal by nanoparticles (NPs) and xerogels (XGs) at ultra-trace levels from seawater samples. Experimental conditions: 10 mL of the seawater solution, with 0.5 Bq/mL for both U-232 and Am-241 radionuclides.

**Figure 11 gels-09-00690-f011:**
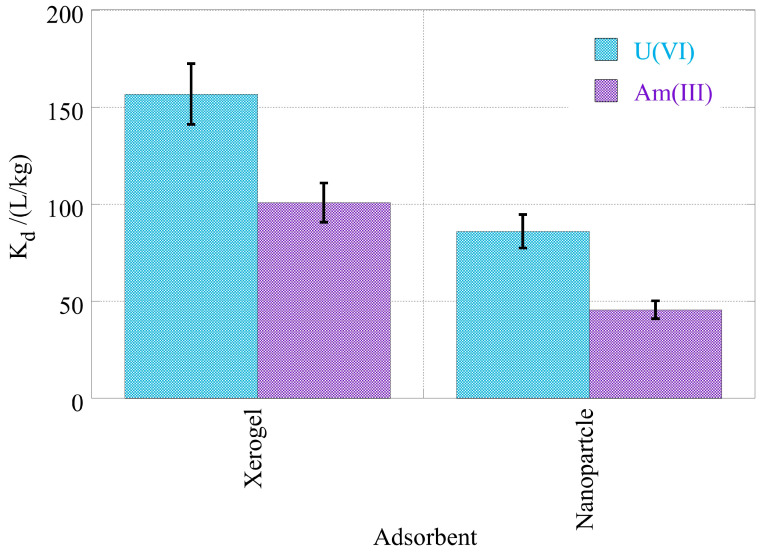
Adsorption efficiency (log_10_K_d_) of U(VI) and Am(III) by nanoparticles (NPs) and xerogels (XGs) at ultra-trace levels from seawater samples. Experimental conditions: 10 mL of the seawater solution, with 0.5 Bq/mL for both U-232 and Am-241 radionuclides.

## Data Availability

Not applicable.

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
