# Peer review of "Actinide Ion (Americium-241 and Uranium-232) Interaction with Hybrid Silica–Hyperbranched Poly(ethylene imine) Nanoparticles and Xerogels"

_gels, 2023, doi:10.3390/gels9090690_

Round 1

Reviewer 1 Report

The manuscript with the title " Actinide ion (Americium-241 and Uranium-232) Interaction with Hybrid Silica-Hyperbranched Poly(ethylene imine) Nano-particles and Xerogels " (gels-2572882) is quite interesting. However, to be published in Gels, this manuscript needs some improvement. Here are some suggestions that need to be considered:

1.           The authors need to state the novelty of this work clearly in the introduction.

2.           In the introduction, clarify the source of Americium (Am-241) in the natural environment and sea water.

3.           Hyperbranched Poly(ethylene imine) nano-particles and xerogels were prepared by changing the composition of PEI. PEI has primary and secondary amine functional groups. But the surface charge of xerogels at pH 4 is negative whereas nanoparticles are positive even at pH 9. How does the pka value of primary and secondary amine functional group of PEI vary with respect to composition?  Carryout detailed investigation and provide explanation to justify this.

4.           Line 126 says “The xerogels have larger BET surfaces and much bigger pores” but the values are not given in the manuscript.

5.           In line 137, “The  of the actinides by the nanoparticles and xerogels is expected to occur via cation exchange between protons and the U(VI) or Am(III) cations and inner-sphere complex formation between the actinide cations and the polyimine moieties.”  Whereas in the line 185” Yet their removal from the aqueous solution proceeds smoothly and is almost complete in the case of nanoparticles that exhibit a higher positive charge. This is a strong indication that adsorption proceeds mainly via the chelation mechanism affording inner-sphere complexes.

Which mechanism involved on the adsorption of U(VI) or Am(III) cations on nanoparticle or xerogels adsorption occur via ion exchange or chelation?. Please refer the following article to know kind of actinides interactions with amine functional group.

Removal of uranium and thorium from aqueous solution by ultrafiltration (UF) and PAMAM dendrimer assisted ultrafiltration (DAUF), Journal of Radioanalytical and Nuclear Chemistry volume 303pages441–450 (2015).

6.           The adsorption need to be performed minimum of 4 to 5 different temperature required for arriving on conclusion whether the process is endothermic or exothermic.  So, The author need to study adsorption study at 55 and 65 0C and linear plots lnkd vs  1/T should be included in the manuscript.

7.           Page 11, line 293 the sentence should be rewritten for clarity  

8.           In conclusion, the last sentence “The simple derivatization of NP and XG to increase the selectivity towards specific actinides and other metal ions along with their  easy  implementation  in  water treatment technologies, could make these materials attractive candidates  for the decontamination of actinide-contaminated waters, including seawaters. But, there is no selectivity studies reported in the present work. The selectivity of nanoparticle or XG towards U(VI) and Am(III) in presence of lanthanides and d-block elements . please refer this article to get the idea about the selectivity studies

Nano-cerium vanadate: A novel inorganic ion exchanger for removal of americium and uranium from simulated aqueous nuclear waste, Journal of Hazardous Materials, Volume 280, 15 September 2014, Pages 63-70.

9.           The author also advised to refer this paper to improve the introduction and scientific explanation of results and discussion.

Research advances in nuclear wastewater treatment using conventional and hybrid technologies: Towards sustainable wastewater reuse and recovery, Journal of Water Process Engineering, Volume 52, April 2023, 103604

10.       There is a typo error in Line number 339 and superscript and subscript errors in line number 211 and 262.

The quality of English is fine but there are few typo errors.

Author Response

Reviewer 1:  Comments and Suggestions for Authors

The manuscript with the title " Actinide ion (Americium-241 and Uranium-232) Interaction with Hybrid Silica-Hyperbranched Poly(ethylene imine) Nano-particles and Xerogels " (gels-2572882) is quite interesting. However, to be published in Gels, this manuscript needs some improvement. Here are some suggestions that need to be considered:

Our response: Dear Reviewer, thank you for the useful comments

  1. The authors need to state the novelty of this work clearly in the introduction.
    Our response: Although, other studies on the removal of uranium, thorium [Ilaiyaraja et al., Removal of uranium and thorium from aqueous solution by ultrafiltration (UF) and PAMAM dendrimer assisted ultrafiltration (DAUF). J Radioanal Nucl Chem 303, 441–450 (2015] and other radionuclides have been reported [Kadadou et al., Research advances in nuclear wastewater treatment using conventional and hybrid technologies: Towards sustainable wastewater reuse and recovery. Journal of Water Process Engineering, 52, 2023, 103604], to the best of our knowledge, this is the first study on hybrid silica-hyperbranched poly(ethylene imine) nanoparticle and xerogel materials used as adsorbents for americium and uranium present in the studied solutions at ultra-trace levels (about nine orders of magnitude lower actinide concentration). We have included also americium in the present study, because americium, is present, like other trans-uranium elements at very low levels, in marine environments, because of global fallout from nuclear bomb tests, accidental releases (e.g. Fukushima accident) and effluent discharges from irradiated fuel reprocessing [Hetherington ey al., in Impacts of nuclear releases into the aquatic environment (IAEA, Vienna) 1975, 193.; Nevissi and Schell, Distribution of plutonium and americium in Bikini Atoll Lagoon. Health Phys., 1975, 28, 539–47.;  Yamamoto Met al., Isotopic Pu, Am and Cm signatures in environmental samples contaminated by the Fukushima Dai-ichi Nuclear Power Plant accident. J Environ Radioact. 2014, 132, 31-46.]. However, studies dealing with americium removal from radioactively contaminated waters are limited [Kadadou et al., Research advances in nuclear wastewater treatment using conventional and hybrid technologies: Towards sustainable wastewater reuse and recovery. Journal of Water Process Engineering, 52, 2023, 103604].

  2. In the introduction, clarify the source of Americium (Am-241) in the natural environment and sea water.
    Our response: Americium, is present, at very low levels, in marine environments, because of global fallout from nuclear bomb tests, accidental releases (e.g. Fukushima accident) and effluent discharges from irradiated fuel reprocessing [Hetherington et al.,. in Impacts of nuclear releases into the aquatic environment (IAEA, Vienna) 1975, 193.; Nevissi and Schell, Distribution of plutonium and americium in Bikini Atoll Lagoon. Health Phys., 1975, 28, 539–47.;  Yamamoto M., et al. Isotopic Pu, Am and Cm signatures in environmental samples contaminated by the Fukushima Dai-ichi Nuclear Power Plant accident. J Environ Radioact. 2014, 132, 31-46]

  3. Hyperbranched Poly(ethylene imine) nano-particles and xerogels were prepared by changing the composition of PEI. PEI has primary and secondary amine functional groups. But the surface charge of xerogels at pH 4 is negative whereas nanoparticles are positive even at pH 9. How does the pka value of primary and secondary amine functional group of PEI vary with respect to composition?  Carryout detailed investigation and provide explanation to justify this.
    Our response: Hyperbranched PEI has primary and secondary and tertiary amino groups amine functional groups. The pKa values do not vary with respect to composition. When the content of Hyperbranched PEI is high the behaviour of the composites approaches the values obtained for the polyamines. On the contrary, when the content of hyperbranched PEI is lower the behaviour of the composites approaches the values obtained for pure silica. A detailed investigation has already been performed and published in a previous work. All this information is now included in the manuscript.

  4. Line 126 says “The xerogels have larger BET surfaces and much bigger pores” but the values are not given in the manuscript.
    Our response: The sentence is corrected and the values are added to the manuscript.

  5. In line 137, “The of the actinides by the nanoparticles and xerogels is expected to occur via cation exchange between protons and the U(VI) or Am(III) cations and inner-sphere complex formation between the actinide cations and the polyimine moieties.”  Whereas in the line 185” Yet their removal from the aqueous solution proceeds smoothly and is almost complete in the case of nanoparticles that exhibit a higher positive charge. This is a strong indication that adsorption proceeds mainly via the chelation mechanism affording inner-sphere complexes.Which mechanism involved on the adsorption of U(VI) or Am(III) cations on nanoparticle or xerogels adsorption occur via ion exchange or chelation?. Please refer the following article to know kind of actinides interactions with amine functional group. Removal of uranium and thorium from aqueous solution by ultrafiltration (UF) and PAMAM dendrimer assisted ultrafiltration (DAUF), Journal of Radioanalytical and Nuclear Chemistry volume 303, pages441–450 (2015).
    Our response: In the present case ion-exchange and inner-sphere complex formation occur successively, particularly in the acidic pH region, where both studied actinides (Am(III) and U(VI)) are expected to be positively charged. Specifically, when the actinide cations approach the protonated imine-groups the proton is exchanged by the actinide ion (ion-exchange), which then is complexed by the imine-groups through the interaction between the amine-lone pair and the empty actinide orbitals. In neutral and alkaline pH region (due to the extremely  low actinide concentration) the actinide complexation may occur directly with the partially de-protonated amino groups [Ilaiyaraja et al. Removal of uranium and thorium from aqueous solution by ultrafiltration (UF) and PAMAM dendrimer assisted ultrafiltration (DAUF). J Radioanal Nucl Chem 303, 441–450 (2015.]. This explanation has been added in the discussion.

  6. The adsorption need to be performed minimum of 4 to 5 different temperature required for arriving on conclusion whether the process is endothermic or exothermic. So, The author need to study adsorption study at 55 and 65 0C and linear plots lnkd vs  1/T should be included in the manuscript.
    Our response: a lnKd vs 1/T has been provided in the supplementary information with additional data obtained at 30 and 40 oC. However, we could not perform any experiments at higher temperatures due to material dissolution.

  7. Page 11, line 293 the sentence should be rewritten for clarity 
    Our response: we have rephrased the sentence

  8. In conclusion, the last sentence “The simple derivatization of NP and XG to increase the selectivity towards specific actinides and other metal ions along with their  easy  implementation  in  water treatment technologies, could make these materials attractive candidates  for the decontamination of actinide-contaminated waters, including seawaters. But, there is no selectivity studies reported in the present work. The selectivity of nanoparticle or XG towards U(VI) and Am(III) in presence of lanthanides and d-block elements . please refer this article to get the idea about the selectivity studies. Nano-cerium vanadate: A novel inorganic ion exchanger for removal of americium and uranium from simulated aqueous nuclear waste, Journal of Hazardous Materials, Volume 280, 15 September 2014, Pages 63-70.
    Our response: we have referred to the above mentioned study.

  9. The author also advised to refer this paper to improve the introduction and scientific explanation of results and discussion. Research advances in nuclear wastewater treatment using conventional and hybrid technologies: Towards sustainable wastewater reuse and recovery, Journal of Water Process Engineering, Volume 52, April 2023, 103604
    Our response: we have referred to the above mentioned paper to improve the introduction and data discussion.
  10. There is a typo error in Line number 339 and superscript and subscript errors in line number 211 and 262.
    Our response: the typo errors have been corrected.

Reviewer 2 Report

The manuscript (gels-2572882) titled with “Actinide ion (Americium-241 and Uranium-232) Interaction with Hybrid Silica-Hyperbranched Poly(ethylene imine) Nanoparticles and Xerogels” investigated the binding of actinide ions (Am(III) and U(VI)) in aqueous solution by hybrid silica-hyperbranched poly (ethylene imine) nanoparticles (NP) and xerogels (XG) at different pH values (4, 7 and 9). The article indicated that the relatively slow adsorption process was governed by the actinide diffusion from the aqueous phase to the solid surface. It was also noted that the favourable effect of the increased temperature was associated with the increasing randomness at the solid-liquid interphase upon actinide adsorption. Compared to other adsorbent materials used for binding Am(III) and U(VI), both materials showed the far higher binding efficiency, which could make these materials attractive candidates for the treatment of radionuclide/actinide contaminated water. However, some important information should be clarified before publication.

The recommendations are given below:

1. In the section of “Introduction”, the authors mentioned that “a few studies that report on the adsorption of these two radionuclides by various adsorbent materials such as microplastics and aerogels.” More references should be cited to support this conclusion.

2. The authors mentioned that “the xerogels have larger BET surfaces and much bigger pores” in line 126-127, however there doesn't seem to be any more evidence of this. Whether the relevant characterisation should be supplemented to more clearly show the difference between Xerogel and Nanoparticle?

3. Although the article explained the differences and reasons for the adsorption effect of the Xerogel and Nanoparticle at different pH, the comparison between the two materials appears to be inadequate, and the relevant explanations are unclear.

4. The authors explained the possible reasons for temperature could affect ΔHo values through changing the "negatively charged species" in 2.4, however were there any other effects of temperature change, such as the surface-active sites mentioned earlier by the authors? More results and references should be added to prove this inference.

5. The innovation of this article were not clear, which should be further emphasized in the part of “Introduction” and “Conclusions”.

6. The authors mentioned that “Compared to other adsorbents, both composites show far higher removal efficiency from laboratory and seawater samples”, yet no clear comparative results were given to justify this conclusion. A comparison of the Am(III) and U(VI) removal performance of composite materials in this work with the reports in the literatures is suggested.

Moderate editing of English language required.

Author Response

Reviewer 2: Comments and Suggestions for Authors

The manuscript (gels-2572882) titled with “Actinide ion (Americium-241 and Uranium-232) Interaction with Hybrid Silica-Hyperbranched Poly(ethylene imine) Nanoparticles and Xerogels” investigated the binding of actinide ions (Am(III) and U(VI)) in aqueous solution by hybrid silica-hyperbranched poly (ethylene imine) nanoparticles (NP) and xerogels (XG) at different pH values (4, 7 and 9). The article indicated that the relatively slow adsorption process was governed by the actinide diffusion from the aqueous phase to the solid surface. It was also noted that the favourable effect of the increased temperature was associated with the increasing randomness at the solid-liquid interphase upon actinide adsorption. Compared to other adsorbent materials used for binding Am(III) and U(VI), both materials showed the far higher binding efficiency, which could make these materials attractive candidates for the treatment of radionuclide/actinide contaminated water. However, some important information should be clarified before publication.

Our response: Dear Reviewer, thank you for the useful comments

The recommendations are given below:

  1. In the section of “Introduction”, the authors mentioned that “a few studies that report on the adsorption of these two radionuclides by various adsorbent materials such as microplastics and aerogels.” More references should be cited to support this conclusion.
    Our response: we have expanded the introduction and included more citations (see also response to reviewer 1 comments)

  2. The authors mentioned that “the xerogels have larger BET surfaces and much bigger pores” in line 126-127, however there doesn't seem to be any more evidence of this. Whether the relevant characterisation should be supplemented to more clearly show the difference between Xerogel and Nanoparticle?
    Our response: The sentence is corrected and the values are added to the manuscript.

  3. Although the article explained the differences and reasons for the adsorption effect of the Xerogel and Nanoparticle at different pH, the comparison between the two materials appears to be inadequate, and the relevant explanations are unclear.
    Our response: Our previous open-access work is entirely dedicated to an extensive comparison of the two adsorbing materials: Arkas, et al. 2023. Comparative Study of the U (VI) Adsorption by Hybrid Silica-Hyperbranched Poly (ethylene imine) Nanoparticles and Xerogels. Nanomaterials13(11), p.1794.

  4. The authors explained the possible reasons for temperature could affect ΔHo values through changing the "negatively charged species" in 2.4, however were there any other effects of temperature change, such as the surface-active sites mentioned earlier by the authors? More results and references should be added to prove this
    Our response: we have included also data obtained for the ΔHo and ΔSo  calculation (see also response to reviewer 1 comments)

  5. The innovation of this article were not clear, which should be further emphasized in the part of “Introduction” and “Conclusions”.
    Our response: we have emphasized the novelty of the present study in the introduction (see also response to reviewer 1 comments)

  6. The authors mentioned that “Compared to other adsorbents, both composites show far higher removal efficiency from laboratory and seawater samples”, yet no clear comparative results were given to justify this conclusion. A comparison of the Am(III) and U(VI) removal performance of composite materials in this work with the reports in the literatures is suggested.
    Our response: Since this study has been performed at ultra-trace levels and in addition studies on the americium adsorption are very limited direct comparison with other studies is difficult. Therefore, we have compared the data of this study with previous studies we have performed under similar conditions with biochar materials and aerogels. In addition, The removal of actinides (including americium and uranium) from low and intermediate active solutions has been studied using inorganic adsorbents/exchangers (e.g. titanosilicates, layered manganese oxides, iron and titanium oxides and nano-cerium vanadate [Rathore, et al., Removal of actinides and fission products activity from intermediate alkaline waste using inorganic exchangers, J. Radioanal. Nucl. Chem. 262 (3) (2004) 543–549.; Al-Attar, et al., Uptake of radionuclides on microporous and layered ion exchange materials, J. Mater. Chem. 13 (2003) 2963–2968.; Lujaniene, et al., Application of inorganic sor- bents for removal of Cs, Sr, Pu and Am from contaminated solutions, J. Radioanal. Nucl. Chem. 282 (2009) 787–791.; Banerjee et al., Nano-cerium vanadate: A novel inorganic ion exchanger for removal of americium and uranium from simulated aqueous nuclear waste. Journal of Hazardous Materials, 280, 2014, 63-70. https://doi.org/10.1016/j.jhazmat.2014.07.026.). The associated adsorption kinetics are significantly faster and the adsorption efficiencies (%-removal and Kd values) remarkably higher. However, both the faster kinetics and the higher adsorption efficiencies could be ascribed to the significantly higher actinide concentrations used in those studies, which more than several orders of magnitude higher than the uranium and americium levels used in the present study.

Round 2

Reviewer 1 Report

The authors answered all the questions and revised the manuscript appropriately. So, the manuscript can be accepted in its current form.

Reviewer 2 Report

The authors have been modified the manuscript, which can be accepted. 

Minor editing of English language required.